# Exploring the Impact of Endometrioma Aspiration and Dienogest Combination Therapy on Cyst Size, Inflammatory Cytokines in Follicular Fluid and Fertility Outcomes

**DOI:** 10.3390/ijms241612891

**Published:** 2023-08-17

**Authors:** Mamoru Shigeta, Isao Tsuji, Shu Hashimoto, Udayanga Sanath Kankanam Gamage, Masaya Yamanaka, Aisaku Fukuda, Yoshiharu Morimoto, Daisuke Tachibana

**Affiliations:** 1IVF Osaka Clinic, Higashi-Osaka, Osaka 577-0012, Japan; shigeta585@ivfjapan.com (M.S.); tsuji545@ivfjapan.com (I.T.); fukuda@ivfosaka.com (A.F.); 2Graduate School of Medicine, Osaka Metropolitan University, Osaka 545-8585, Japan; 3HORAC Grand Front Osaka Clinic, Osaka 530-0011, Japan; udayanga646@ivfjapan.com (U.S.K.G.); yamanaka086@ivfjapan.com (M.Y.); ivfceo@gmail.com (Y.M.); 4Women’s Lifecare Medicine, Obstetrics and Gynecology, School of Medicine, Osaka Metropolitan University, Osaka 545-8585, Japan; dtachibana@omu.ac.jp

**Keywords:** endometriosis, dienogest, cytokines, endometrioma aspiration, controlled ovarian stimulation

## Abstract

Endometriomas (chocolate cysts) are cystic lesions that can develop on ovaries, and are characterized by the presence of ectopic endometrial tissue or similar tissue. Such lesions can cause a decline in the number and quality of oocytes, and lead to implantation failure. In this study, we retrospectively assessed the efficacy of repeated endometrioma aspiration and dienogest combination therapy in patients suffering endometriosis-associated infertility with endometriomas. A comparison was made between a treated group that underwent combination therapy followed by controlled ovarian hyperstimulation (COH) (*n* = 30) and a control group that did not undergo treatment (*n* = 40), at the IVF Osaka Clinic from September 2019 to September 2021. There were no differences in patient background between the two groups. A reduction in endometrioma size continued for 12 months after treatment. The numbers of follicles that developed to 15 mm or greater in size following COH and mature oocytes were significantly lower in the treated group compared to those in the control group. The levels of inflammatory cytokines in the follicular fluid significantly decreased in the treated group (*p* < 0.05). In patients in the treated group who underwent a second ova retrieval, the results were compared between those in the first ova retrieval (immediately after the end of treatment) and those in the second ova retrieval (four months after the first retrieval). The numbers of follicles following COH, retrieved, mature and fertilized ova were significantly increased in the second ova retrieval.

## 1. Introduction

Endometriosis is a condition in which tissue similar to the lining of the uterus, called the endometrium, grows outside the uterus. Endometriosis is an estrogen-dependent disease associated with symptoms such as painful menstruation (dysmenorrhea), chronic pelvic pain and infertility [1,2]. This disease affects approximately 10% of women of reproductive age, and is even more prevalent among women with infertility, affecting 20–50% of that group [1,2]. One common manifestation of endometriosis is the presence of endometriomas, which are cystic or tumor-like growths filled with endometrial tissue that develop on the ovaries. These endometriomas are found in approximately 30% of women with endometriosis [3]. The clinical outcomes of assisted reproductive technology (ART) in patients with endometriosis are inferior to those among the general infertile female population [4]. Endometriomas, along with the inflammatory environment they create, release reactive oxygen species (ROS) and inflammatory cytokines that have a negative impact on the surrounding ovarian environment [5,6]. These impacts can lead to a decrease in the number and quality of oocytes due to their detrimental effects on the follicular fluid and cumulus cells [5,7,8]. Additionally, the eutopic endometrium (the normal endometrial tissue in the uterus) of women with endometriosis is often resistant to the effects of progesterone, which is necessary for proper endometrial preparation and decidualization, further contributing to implantation failure [9,10].

Surgical treatment options, such as cystectomy (removal of endometriomas), have been used in infertile patients with endometriomas. However, studies have shown that these procedures do not improve the outcomes of ART [11], and may even lead to a decrease in the ovarian reserve, which refers to the number and quality of eggs remaining in the ovaries. As a result, if the size of the endometrioma is small, surgical treatment is typically avoided. However, there are concerns associated with not removing endometriomas, including adverse effects on the eggs, infection during ova pick-up (OPU), contamination of follicular fluid with endometrial contents, obstetric complications such as an endometrioma rupture during pregnancy, and the potential risk of the endometriomas transforming into malignant tumors [12]. On the other hand, performing a cystectomy to remove endometriomas can lead to the removal of normal ovarian tissue along with the endometriomas, resulting in a decrease in ovarian reserves [13]. Notably, bilateral removal of endometriomas was associated with a 2.4% risk of premature postoperative ovarian insufficiency [14]. Additionally, even without undergoing cystectomy, patients with endometriomas may experience reduced ovarian reserves due to the effects of ROS and cytokines released by the endometriomas [5,15]. Therefore, in cases where patients have bilateral endometriomas or already diminished ovarian reserves before surgery, alternative treatments other than cystectomy may need to be considered. In the field of reproductive medicine, preserving fertility such as cryopreservation of oocytes, embryos or ovarian tissue [16], and improving pregnancy rates, are primary challenges for infertile patients with endometriomas.

One alternative to cystectomy is the aspiration of endometriomas using transvaginal ultrasound guidance. This procedure is minimally invasive, simple [17], and helps preserve ovarian reserves [18]. Although the recurrence of endometriomas is a common drawback of aspiration, studies have shown that combining the use of gonadotropin-releasing hormone (GnRH) agonists and multiple aspirations can reduce the frequency of recurrence and improve ART outcomes [19,20]. Hormonal therapies, such as GnRH agonists [21,22] and the progestin dienogest (DNG) [23] administered prior to ART have also been reported to enhance clinical pregnancy rates, but recent systematic reviews have debunked the effects of GnRH agonists [24] and DNG [25], suggesting that it is difficult to expect that drugs alone can improve ART outcomes in endometriosis patients.

Combining DNG therapy and multiple aspirations before ART has the potential to improve ART outcomes compared to a combination of GnRH agonists and multiple aspirations, because DNG therapy has been reported to increase the expression of progesterone receptors [26] and reduce the size of endometriomas in patients with endometriosis. However, to date, there have been no studies conducted to specifically investigate the effects of combining endometrioma aspiration with DNG therapy prior to ART in patients with endometrioma-associated infertility.

Therefore, the objective of this study was to retrospectively evaluate the effectiveness of this treatment approach in reducing the size of endometriomas and improving the outcomes of ART in infertile patients with endometriomas.

## 2. Results

### 2.1. Patient Characteristics

Overall, this study included 70 patients, 30 of whom received endometrioma aspiration combined with DNG therapy before IVF, and 40 of whom underwent IVF without previous hormone or aspiration therapy (Figure 1). The characteristics of the patients in each group are described in Table 1. The mean diameter of the largest endometrioma at pre-examination in the treated group was 43.8 ± 13.0 mm, which was significantly larger than the control group mean diameter of 29.8 ± 10.3 mm (*p* < 0.05). In all cases, OPU was performed following COH.

### 2.2. Effect on the Size of Endometriomas

In the treated group, the changes in endometrioma size before and after endometrioma aspiration combined with DNG therapy were investigated. Among the 30 patients in the treated group, the majority underwent endometrioma aspiration three times (24 patients), while some underwent the procedure twice (4 patients) or once (2 patients).

The results showed a significant decrease in the diameter of the endometrioma over time. The pre-treatment measurement of 43.8 ± 13.0 mm decreased to 20.4 ± 8.30 mm at 0.5 months (*p* < 0.05), 20.8 ± 7.51 mm at 1 month (*p* < 0.05), 22.2 ± 6.17 mm at 3 months (*p* < 0.05), 22.8 ± 6.69 mm at 6 months (*p* < 0.05), 20.8 ± 7.15 mm at 9 months (*p* < 0.05) and 23.6 ± 6.91 mm at 12 months (*p* < 0.05) after completion of the treatment (Figure 2).

Notably, cases of pregnancy that occurred during the study period, as well as cases of treatment withdrawal upon the patient’s request, were excluded from the analysis to focus on the treatment outcomes. In the treated group, no cases of infection or ruptures of the endometriomas occurred after the aspiration procedure. Additionally, no cases showed malignant findings in the cytology examination performed at the time of aspiration. These findings suggest that endometrioma aspiration combined with DNG therapy effectively reduced the sizes of the endometriomas without adverse events such as infection or rupture.

### 2.3. Clinical Outcomes of IVF–ET in the Control and Treated Groups

The outcomes of the control and treated groups are summarized in Table 2. The number of follicles that grew to 15 mm or more on transvaginal ultrasonography following COH was 6.6 ± 3.7 in the treated group, which was significantly lower than that in the control group (10.5 ± 6.4, *p* < 0.05). On the trigger day, the serum E_2_ level in the treated group was 1948 ± 1072 ng/mL, which was significantly lower than the level observed in the control group (2564 ± 1232 ng/mL, *p* < 0.05). However, there was no significant difference in the dosage or duration of FSH between the two groups. There was no significant difference in the numbers of ova retrieved between the two groups. The number of mature oocytes (8.1 ± 6.2) in the treated group was significantly lower than that in the control (11.5 ± 7.4, *p* < 0.05), although there was no statistically significant difference between two groups in maturation rate, the number of fertilized ova, the fertilization rate, the number of blastocysts, the blastulation rate, the number of morphologically good blastocysts, the morphologically good blastocyst rate, the implantation rate, the pregnancy rate or the live birth rate. Overall, while the treated group presented lower numbers of mature oocytes and a lower serum E_2_ level, there were no significant differences in the fertilization outcomes, implantation rate, pregnancy rate or live birth rate compared to the control group.

### 2.4. Inflammatory Cytokine Levels in Follicular Fluid

Follicular fluid was collected from follicles that had developed to at least 18 mm in diameter, and the levels of inflammatory cytokines in the follicular fluid were measured (Figure 3). The follicular fluid from the left and right sides of each patient was collected separately, and the mean value of the left and right sides was used as the value for that patient. We observed a trend toward lower levels of inflammatory cytokines in follicular fluid in the treated group. In particular, IL-6 (control: 56.9 ± 82.9 vs. treated: 25.6 ± 29.9 pg/mL), IL-8 (control: 818.6 ± 583.2 vs. treated: 664.0 ± 723.5 pg/mL), IL-10 (control: 18.4 ± 22.9 vs. treated: 6.8 ± 7.9 pg/mL), IL-18 (control: 795.9 ± 593.0 vs. treated: 515.0 ± 349.6 pg/mL), IL-23 (control: 9.7 ± 10.3 vs. treated: 3.1 ± 2.6 pg/mL), IFN-α2 (control: 9.3 ± 13.5 vs. treated: 3.1 ± 3.5 pg/mL), and MCP-1 (control: 1305 ± 1000 vs. treated: 871.2 ± 451.5 pg/mL) were significantly lower in the treated group (*p* < 0.05).

### 2.5. Effect of Time after Treatment on Clinical Outcomes

Of the 30 patients in the treated group, 14 patients were detected with heart beat and ongoing pregnancy in results of the first OPU. Out of the remaining 16 patients, 13 underwent a second round of OPU (3 patients withdrew) and 10 of them met the criteria for poor ovarian response (POR) based on the POSEIDON criteria [27] in the first OPU. Nine patients received a second OPU with COH protocol, and one received mild stimulation protocol. To assess ovarian stimulation responsiveness in patients with POR, the results of the first and second retrievals were compared among these nine patients (Table 3). The mean age of these patients before DNG administration was 35.4 ± 3.5 years, and the mean AMH level was 1.6 ± 0.73. The mean interval between the first and second OPU was 4.0 ± 1.1 months. There were significant differences (*p* < 0.05) in the mean number of antral follicles (first round: 4.8 vs. second round: 8.7), the mean number of follicles that reached a size of 15 mm or more (first round: 3.6 vs. second round: 7.8), the serum E_2_ level on the trigger day (first round: 1333 ng/mL vs. second round: 2289), the number of oocytes retrieved (first round: 4.7 vs. second round: 9.3), the number of mature oocytes (first round: 3.7 vs. second round: 7.0), and the number of fertilized oocytes (1.8 vs. 5.6). There were no differences in the maturation rate or fertilization rate. The implantation rates for embryos obtained from the first and second OPU were 25.0% and 40.0%, respectively; the pregnancy rates were 22.2% and 55.6%, respectively; and the live birth rates were 0% and 33.3%, respectively. Although there was a tendency toward positive outcomes, no statistically significant differences were observed using Fisher’s exact test. Overall, the second OPU in poor responders to the first OPU showed improvements in the number of antral follicles, mature follicles, retrieved oocytes, mature oocytes and fertilized oocytes. Although positive outcomes were observed in terms of implantation, pregnancy and live birth rates, the differences did not reach statistical significance in this limited sample size analysis.

## 3. Discussion

The results of the present study demonstrated several important findings. Firstly, the combination of endometrioma aspiration and DNG therapy led to a long-term reduction in the sizes of the endometriomas. This result indicates that the treatment approach was effective in shrinking the endometriomas over time. Additionally, the study showed a decrease in inflammatory cytokines in the follicular fluid following treatment. This result suggests that the combination therapy had a positive impact on the inflammatory environment associated with endometriomas. However, when comparing the immediate clinical outcomes of ART between the treated group and the control group (where endometriomas were not treated and only COH was performed), the results were less favorable in the treated group. This result suggests that the immediate response of the treated group to ART was not as successful as that of the control group. Nevertheless, when the OPU was repeated in the same patients approximately four months after treatment, a significant improvement in outcomes was observed compared to the OPU performed immediately after treatment. This result indicates that there was a delayed positive effect of the combination therapy on the ART outcomes in the treated group.

Previously, fourth-generation progestin DNG was shown to attenuate endometriosis, but its effects on ovarian reserves and fertility treatment outcomes have not been fully tested. In a domestic long-term study with DNG in Japan [28], after 108 patients received DNG 2 mg/day for 52 weeks, the proportion of patients with ovarian endometriomas exhibiting shrinkage of 25% or more was 76.9% (83/108 cases) and 84.7% (83/98 cases) at 24 and 52 weeks of treatment, respectively. The remaining 15 patients presented an unchanged or increased cyst size. When DNG treatment was combined with endometrioma aspiration in the present study, cyst reduction of ≥50% was observed in 28 out of 30 patients (93.4%). Cyst shrinkages of <50% and ≥25% were observed in one patient (3.3%), and cyst shrinkages of <25% and ≥10% were observed in one patient (3.3%) (Figure 2b). Interestingly, these results were achieved despite a shorter treatment period of 90 days compared to the 52-week (364 days) treatment period in the previous study [28]. Additionally, the shrinkage effect was also maintained one year after treatment (Figure 2a). These results demonstrate that the addition of endometrioma aspiration to DNG therapy provides a greater short-term reduction in endometrioma size compared to DNG therapy alone. Furthermore, the long-term cyst reduction effect led to the avoidance of cystectomy in all cases, which provides remarkable benefits to the patients. This reduction effect persisted even after the completion of treatment, and within 12 months of treatment, 15 out of 30 patients (50.0%) achieved pregnancy and subsequently gave birth to healthy infants. Endometrioma aspiration does not affect ovarian reserve [18], thereby avoiding the postoperative decline in ovarian reserves that is experienced with procedures like a cystectomy. Therefore, a combination of endometrioma aspiration with DNG therapy may represent an alternative option to avoid cystectomies for patients with already decreased ovarian reserves. However, when performing endometrioma aspiration, it is important to be aware of the potential risks of pelvic peritonitis, ovarian abscess formation, cyst rupture and inadvertent puncturing of ovarian carcinomas.

The reported incidence of pelvic infection is 0–1.4% [29,30], and the risk of concurrent carcinoma in endometriomas is estimated to be 0.9% [31]. In this study, no such complications occurred due to the measures taken, such as administering antibiotics before aspiration, and performing an MRI examination when a malignancy could not be ruled out based on a pre-aspiration ultrasound. However, considering that the combination of endometrioma aspiration with DNG therapy is not a standard treatment for endometriomas, further investigation is needed to evaluate the safety of this method.

In patients with endometriosis, the total number of intra-abdominal macrophages was increased, as were the levels of various cytokines derived from macrophages and endometriotic tissues [32,33,34]. Inflammatory cytokines and ROS in cyst contents, follicular fluid, blood and ascites fluid were also found to be elevated in patients with endometriomas [5,6,35,36]. Increased inflammatory cytokines and ROS in follicular fluid were shown to cause a decrease in the function of the cumulus cells, and a decrease in the number and quality of oocytes [5,6]. In the present study, IL-6, IL-8, IL-10, IL-18, IL-23, IFN-α2 and MCP-1, which are associated with endometriosis and considered to be exacerbating factors in endometriosis, were significantly lower in the treated group (Figure 3). In a previous randomized controlled trial, endometriomas patients treated with DNG therapy alone for three months followed by COH failed to demonstrate a decrease in the inflammatory cytokine levels in follicular fluid compared to a control group who underwent COH without DNG [37]. Hence, a combination of DNG therapy and endometrioma aspiration is likely to result in lower cytokine concentrations in the follicular fluid compared to DNG therapy alone. These reductions in cytokine concentrations in the follicular fluid reflected control of the lesion through shrinkage of endometriomas, and predicted the effectiveness of endometrioma aspiration in combination with DNG therapy in improving the oocyte developmental environment. 

As combination therapy reduced the endometrioma size and inflammatory cytokine levels in follicular fluid, we anticipated an improvement in ART outcomes. However, in the treated group, there was a significant reduction in the numbers of follicles that grew to 15 mm or larger and retrieved mature oocytes. The serum E_2_ level on the trigger day was also significantly reduced compared to the control group. There were no improvements observed in other parameters related to OPU and ET. DNG is known to cause apoptosis of granulosa cells, leading to follicle atresia [38]. Therefore, the reduced number of mature oocytes in the treated groups was considered to be due to the adverse effects of DNG. It was also shown that the anti-androgenic effects of DNG inhibit the activity of phospho-v-akt murine thymoma viral oncogene homologue 1, and inhibit the development of primordial follicles to primary follicles in mice [39]. This mechanism may have reduced the number of developing follicles in the treated group, resulting in a reduction in the number of retrieved mature oocytes.

It was demonstrated in human studies that administering DNG followed immediately by a combination of estrogen and progesterone (EP) to induce withdrawal bleeding, and initiation of ovarian stimulation can reduce the number of retrieved, mature and fertilized ova, as well as the rates of fertilization, implantation, clinical pregnancies and live births [37]. In this study, to ensure a preparation period of one menstrual cycle, the patients waited for spontaneous ovulation after completion of DNG administration, and EP administration was started after spontaneous ovulation. However, the number of retrieved mature oocytes was significantly reduced in the treated group. Considering that it takes approximately six months from primordial follicle emergence to ovulation, there is a possibility that the adverse effects of DNG on oocytes persisted even during the one-month menstrual cycle preparation period.

In the treated group, a comparison was made between the outcomes of the first and second OPU in nine cases that presented POR in the first OPU and underwent the second OPU with COH protocol, mainly to evaluate the ovarian response (Table 3). The mean interval between the first and second OPU was 4.0 ± 1.1 months. In the second OPU, the AFC, the serum E_2_ on the trigger day, the number of oocytes retrieved, mature oocytes and fertilized oocytes significantly increased, indicating improved OPU performance (Table 3). Of these nine cases, the fertilized ova from the second OPU resulted in five cases of pregnancy and three live births. The significant increase in the number of antral follicles in the second OPU indicates that the inhibitory effects of DNG on the development of primordial follicles may have resulted in a decreased number of antral follicles and a consequent decrease in the number of retrieved oocytes, even with a post-treatment preparation period (in the present study, the period from the end of DNG to the start of withdrawal bleeding). These results indicate that a preparation period of one menstrual cycle following DNG administration is insufficient. A longer preparation period may improve the development of follicles and potentially lead to an increase in the number of retrieved oocytes, mature oocytes and fertilized oocytes.

This study has several limitations, including its retrospective analysis and small sample size. Furthermore, endometrioma aspiration combined with DNG therapy is not a standard treatment for endometriomas. There are potential risks associated with this approach, such as pelvic peritonitis, ovarian abscess, cyst rupture and the accidental puncture of ovarian carcinoma. To confirm the effectiveness of this treatment, larger prospective studies with careful consideration of safety factors are needed.

## 4. Materials and Methods

### 4.1. Study Design and Patient Selection

The study population consisted of infertile patients with endometriomas who sought treatment at the IVF Osaka Clinic in Osaka, Japan, between September 2019 and September 2021. The aim of the study was to evaluate the effects of endometrioma aspiration combined with DNG therapy. A retrospective analysis was conducted to assess the sizes of endometriomas, clinical outcomes of ART, and concentrations of inflammatory cytokines in follicular fluid. The study included a treated group that underwent endometrioma aspiration combined with DNG therapy prior to controlled ovarian hyperstimulation (COH), and a control group that underwent COH for only endometriomas. Ovarian pick-up (OPU) was performed following COH.

The study protocol was approved by the Institutional Review Board of IVF Osaka Clinic and Osaka City University, Osaka, Japan. The study was conducted in accordance with the principles outlined in the Declaration of Helsinki and the International Conference on Harmonization Guidelines for Good Clinical Practice. Written informed consent was obtained from all enrolled patients.

The eligibility criteria for the study participants were defined as follows: (1) infertility complicated due to endometriomas, (2) scheduled ART with COH, and (3) an age under 40 years. Patients with a history of abdominal surgery or hormone therapy within 12 months of OPU, aspiration to ovarian cysts or tumors, pelvic inflammatory disease, autoimmune disorders, any other immune-affecting exposures, or polycystic ovary syndrome were excluded from this study. Among the eligible patients during the study period, 30 patients who were scheduled to undergo endometrioma aspiration combined with DNG therapy were assigned to the treated group, while 40 patients who did not receive this treatment were assigned to the control group, where they underwent ART without any specific treatment for endometriomas.

### 4.2. The Procedure of Endometrioma Aspiration Combined with DNG Therapy

The treatment protocol for the treated and control groups is presented in Figure 4. The treated group received oral administration of DNG (dienogest, 2 mg/day; Mochida Pharmaceutical Co., Ltd., Tokyo, Japan) for a duration of 90 days starting on the 5th day of the menstrual cycle. Initial endometrioma aspiration was performed on the 5th day after the patient began taking DNG. Subsequent endometrioma aspirations were conducted on the 40th and 80th days after initiating DNG treatment if transvaginal ultrasound examination revealed a recurrence of endometriomas larger than 10 mm. Transvaginal puncture of the endometriomas was performed using a 17 G needle under intravenous anesthesia, guided by transvaginal ultrasound. Patients received preoperative 500 mg ceftriaxone (rocephin for injection, TAIYO Pharma Co., Tokyo, Japan) intravenously on the day of puncture, and cefaclor 250 mg (Kefral capsules, KYOWA Pharma Co., Osaka, Japan) three times daily for two days starting the next day. The contents of the endometriomas were aspirated, and the endometriomas were rinsed with warm saline solution. Only cases in which malignancy was ruled out by contrast-enhanced MRI prior to treatment were included. Cytology of the aspirated contents was performed in all cases of endometrioma aspiration to confirm the absence of malignant cells.

### 4.3. COH, In Vitro Fertilization (IVF) and Embryo Transfer (ET)

In the treated group, after the first ovulation following endometrioma aspiration combined with DNG therapy, ethinylestradiol–norgestrel (Planovar tablets, 1 tablet/day, ASKA Pharmaceutical Co., Tokyo, Japan) was administered for 10 days to induce withdrawal bleeding. From the subsequent menstrual cycle, controlled ovarian hyperstimulation (COH) was initiated using one of the following protocols: short, long or antagonist protocol.

In the short protocol, a nasal spray GnRH agonist (Suprecur: buserelin acetate, 900 μg/day; Mochida Pharmaceutical Co., Tokyo, Japan) was administered daily, starting from the second day of the OPU cycle. This medication helps suppress pituitary gonadotropin secretion continuously until the injection of recombinant human chorionic gonadotropin (rHCG, 250 μg; Ovidrel, Merck Biopharma Co., Ltd., Tokyo, Japan) for ovulation induction. From the third day of the OPU cycle until the day before rHCG injection, a daily injection of 225 IU of recombinant human follicle-stimulating hormone (rFSH; Gonal-F, Merck Biopharma Co., Ltd., Tokyo, Japan) was given. 

In the long protocol, a nasal spray GnRH agonist was administered from the mid-luteal phase in the previous cycle of the OPU until the time of rHCG injection for ovulation induction. From the third day of the IVF cycle until the day before rHCG injection, a daily injection of 225 IU of rFSH was given.

In the antagonist protocol, daily injections of 225 IU of rFSH were initiated from the third day of the IVF cycle until the day before the rHCG injection. Daily administration of a GnRH antagonist (cetrorelix 0.25 mg; Cetrotide, Merck Biopharma Co., Ltd., Tokyo, Japan) started when the leading follicle reached at least 14 mm in diameter and continued until the day before rHCG injection.

In all of the protocols, if the follicle growth rate was assessed as slow, the rFSH dose was increased by 75 IU. Final oocyte maturation was induced by a single administration of 250 μg of rHCG when two or more leading follicles reached a diameter of 18 mm or more. After 35 h of rHCG administration, transvaginal ultrasound-guided OPU was performed using an 18 G needle under intravenous anesthesia. All follicles with a mean diameter of 10 mm or greater were aspirated to collect oocytes.

The protocol for OPU was determined based on the patient’s anti-Müllerian hormone (AMH) and antral follicle count (AFC). The antagonist protocol was employed for cases with AMH ≥ 4.0 or AFC ≥ 15. The long protocol was employed for cases with AMH < 4 and AFC < 15 but ≥7. For cases with AMH < 4 and AFC < 7, either the short or antagonist protocol was employed.

The retrieved cumulus–oocyte complexes were fertilized by either conventional insemination or intracytoplasmic sperm injection, depending on the sperm quality. After OPU, fresh embryo transfer (ET) was carried out on either day 3 or 5 of the in vitro culture. In cases where the endometrium was not in good condition, the patient experienced ovarian hyperstimulation syndrome, or if the patient did not wish to have a fresh ET, embryo freezing was performed. The remaining embryos were vitrified and stored at a low temperature for future use. In a subsequent cycle, the vitrified embryos were warmed and thawed before being transferred into the uterus.

The embryos were evaluated and classified based on specific criteria. On day 3, the embryos were classified according to the criteria proposed by Veeck [40]. A transferable embryo was defined as having at least five blastomeres and less than 25% of its volume filled with fragments.

On day 5, the blastocysts were classified according to the criteria proposed by Gardner [41]. A good quality blastocyst was defined as having a minimum score of 3BB, but not grade C.

Two weeks after embryo transfer, a chemical pregnancy test was performed to determine if implantation had occurred. Subsequently, pregnancy was confirmed when a gestational sac was detected on transvaginal ultrasound three weeks after embryo transfer.

### 4.4. Detection and Diagnosis of Endometriomas

In the present study, the diagnosis of endometriomas was primarily achieved through ultrasonography or MRI rather than surgical findings or histopathological examination. Surgical removal of endometriomas had only been performed in six patients, as indicated in Table 1. However, for all cases of endometrioma aspiration, a visual and cytological examination of the aspirated contents was carried out to ensure consistency with endometrioma. This approach allowed us to confirm the presence of endometriomas and guide the treatment decisions for the patients.

### 4.5. Measurement of Endometrioma Size and Calculation of the Volume

The maximum diameter (D1) of an endometrioma on a section was measured together with the diameter perpendicular to D1 (D2). The mean of D1 and D2 was calculated, and this mean value represented the size of the endometrioma. The volume of the endometrioma was calculated using the following formula: [(D1 + D2) × 0.50]3 × 0.50. According to the changes in endometrioma volume from baseline to the end of endometrioma aspiration combined with DNG therapy (evaluated at 0.5 months after the end of DNG), the patients were divided into four groups: significantly reduced (≥ 50%), reduced (<50%, ≥25%), slightly reduced (<25%, ≥10%) and unchanged (<10%) or increased. The number of patients in each group was counted.

### 4.6. Assessment Items

The endometrioma reduction effect of endometrioma aspiration combined with DNG therapy was evaluated via transvaginal ultrasonography at 0.5, 1, 3, 6 and 12 months after the end of DNG administration. The outcome of OPU was evaluated in terms of the duration and total FSH dose administered for COH, the number of follicles that grew to 15 mm or more in diameter, the serum E_2_ level (estradiol) on the trigger day, the number of retrieved ova, the number of mature ova, the maturation rate, the number of fertilized ova, the fertilization rate, the blastulation rate and the morphologically good blastocyst rate. The outcome of ET was evaluated in terms of implantation rate, pregnancy rate (confirmation of gestational sac) and live birth rate. In patients who consented to the collection and analysis of follicular fluid at the time of OPU (20 in the treated group and 27 in the control group), follicular fluid was collected at the time of OPU and analyzed for interleukin-1β (IL-1β), IL-6, IL-8, IL-10, IL-18, IL-23, interferon-α2 (IFN-α2), IFN-γ, monocyte chemotactic protein-1 (MCP-1) and tumor necrosis factor α (TNF-α), which were measured using a cell sorter as described below.

### 4.7. Collection of Follicular Fluid and Measurement of Cytokines and Chemokines

Follicular fluid of follicles with a diameter of 18 mm or greater was collected under ultrasound guidance from all patients separately in the left and right ovaries (one specimen from each side) at the same OPU time. The follicular fluids containing the oocyte were collected, and immediately after removal of the oocyte, each of the follicular fluid samples was centrifuged at 1200× *g* for 10 min to remove cellular components. The supernatants were stored at −80 °C until they were assayed. The serum cytokine levels were analyzed via multiplex immunoassays and Luminex bead-based antibody capture and recognition arrays (LEGENDplex™, BioLegend, San Diego, CA, USA). Undiluted samples from follicular fluid were run using a cell sorter (SH800, Sony Corporation, Tokyo, Japan). The data were collected with LEGENDplex v8.0 Software. The concentrations of IL-1β, IL-6, IL-8, IL-10, IL-18, IL-23, IFN-α2, IFN-γ, MCP-1 and TNF-α were measured according to the assay protocol.

### 4.8. Statistical Analyses

In this study, statistical analyses were conducted to analyze the data obtained from different measurements and comparisons. Each patient’s basic characteristics and OPU data between the control and treated groups were analyzed using unpaired Student’s *t*-tests, which compare the means of two independent groups. The post-implantation clinical outcome data between the two groups were analyzed using a Pearson chi-squared test, which examines the association between categorical variables. The changes in the size of endometriomas in the treated group and comparison of the first and second OPU results for poor responders in the treated group were analyzed using paired Student’s *t*-tests. These tests compare the means of two related groups. Since the available sample size was limited for the comparison of post-implantation clinical outcomes between the first and second OPU results, Fisher’s exact test, which is used for small sample sizes, was employed. The follicular fluid cytokine levels were analyzed using a Mann–Whitney U test, a nonparametric test that compares two independent groups. *p* < 0.05 was considered statistically significant, indicating that the observed results were unlikely to occur by chance alone. All statistical analyses were performed using GraphPad Prism 5.03 software, a statistical analysis and graphing tool (GraphPad Inc., San Diego, CA, USA).

## 5. Conclusions

The findings of this study suggest that endometrioma aspiration combined with DNG therapy can be an effective treatment option for shrinking endometriomas and reducing inflammatory cytokine levels in follicular fluid. The observed reduction in endometrioma size lasted for up to 12 months after completion of the treatment, indicating a sustained effect. Furthermore, this study suggests that a longer preparation period between the completion of the treatment and the initiation of COH for ART may improve the development of follicles. These results suggest that a single menstrual cycle may not be sufficient for optimal preparation, and that extending the preparation period may help mitigate negative effects on OPU outcomes. This finding emphasizes the need for careful timing and planning in order to optimize ART outcomes following combination therapy. Notably, multiple endometrioma aspiration is not a standard treatment for endometriomas, and the decision to perform this procedure should be made after careful consideration of safety and individual patient indications. Each case should be evaluated on an individual basis, taking into account factors such as ovarian reserves and the patient’s overall reproductive goals. Further research and discussion among healthcare professionals are necessary to determine the safety, efficacy and appropriate indications for multiple endometrioma aspirations in the management of endometriomas.

## Figures and Tables

**Figure 1 ijms-24-12891-f001:**
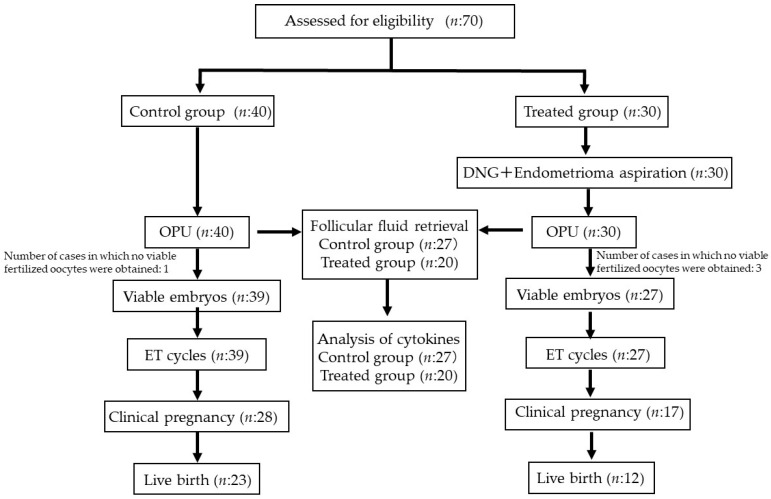
Flow chart of the control and treated groups.

**Figure 2 ijms-24-12891-f002:**
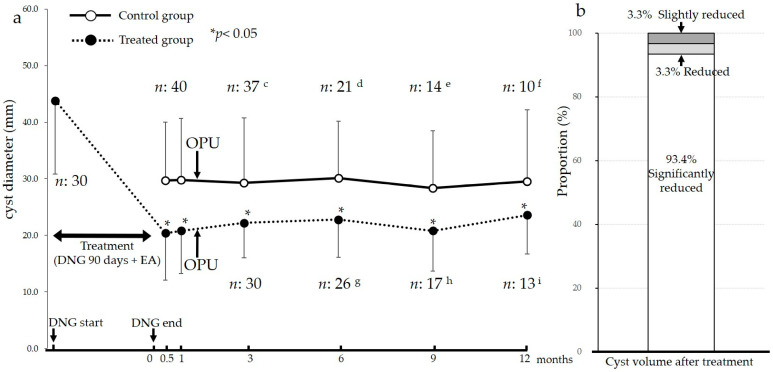
Changes in the sizes of endometriomas in the control and treated groups. This study measured the sizes of endometriomas ((**a**) diameter, (**b**) volume) before and after treatment in the treated group, and from the menstrual cycle prior to ova pick-up (OPU) until 12 months later in the control group. (**b**) The degree of volume reduction was assessed by comparing the measurements before treatment and 0.5 months after treatment. The results showed a significant difference in the sizes of endometriomas before and after treatment at 0.5, 1, 3, 6, 9 and 12 months in the treated group. No significant change in size was observed in the control group. The number of dropouts and their reasons are shown below: (c) *n*:3, withdrawal by patient’s request; (d) n:1, withdrawal by patient’s request, *n*:15, transfer to obstetrics after 11 weeks of pregnancy; (e) *n*:7, transfer to obstetrics after 11 weeks of pregnancy; (f) *n*:4, transfer to obstetrics after 11 weeks of pregnancy; (g) *n*:4, transfer to obstetrics after 11 weeks of pregnancy; (h) *n*:3, withdrawal by patient’s request, *n*:6, transfer to obstetrics after 11 weeks of pregnancy; (i) *n*:4, transfer to obstetrics after 11 weeks of pregnancy. DNG: dienogest, EA: endometrioma aspiration.

**Figure 3 ijms-24-12891-f003:**
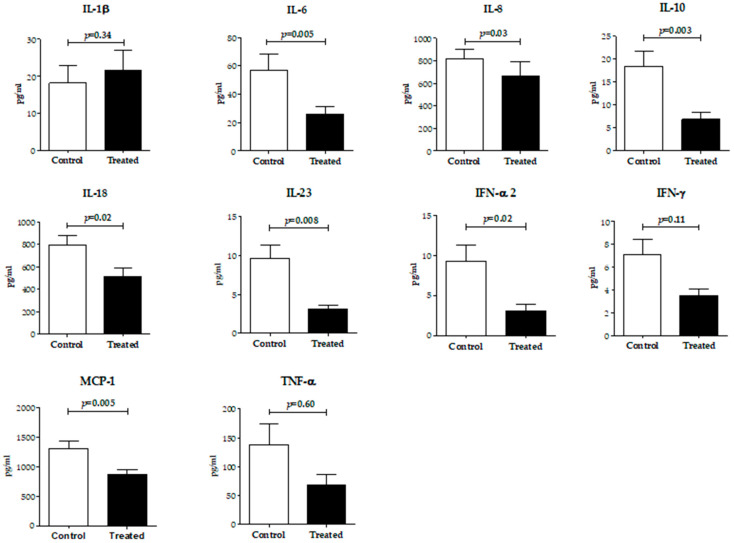
Concentrations of inflammatory cytokines in follicular fluids. In the control group (*n* = 27) and the treated group (*n* = 20), follicular fluids were collected at the time of OPU. Interleukin-1β (IL-1β), IL-6, IL-8, IL-10, IL-18, IL-23, interferon-α2 (IFN-α2), IFN-γ, monocyte chemotactic protein-1 (MCP-1) and tumor necrosis factor α (TNF-α) were analyzed using a cell sorter. Values are presented as mean ± SD. Statistical analyses were performed using a Mann–Whitney U-test.

**Figure 4 ijms-24-12891-f004:**
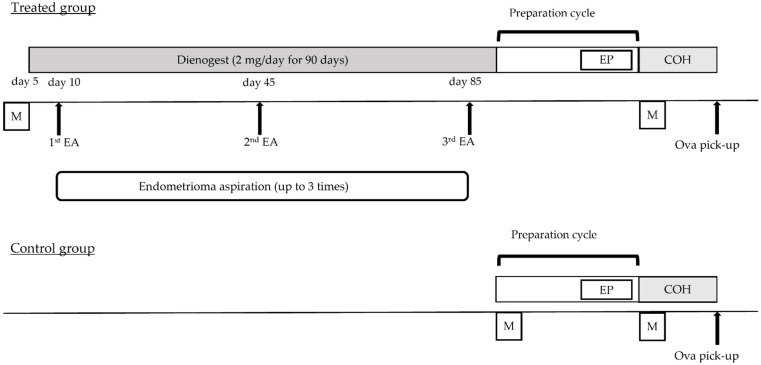
Schematic representation of the study protocol. The treated group received dienogest (DNG) orally for 90 days starting from day 5, and endometrioma aspiration (EA) was performed on day 10. EA was performed again on the 40th and 80th days after the patient started taking DNG if the diameters of endometriomas reached more than 10 mm. Ethinylestradiol–norgestrel (EP) tablets were administered for 10 days (1 tablet/day) after the first ovulation following DNG and EA to induce withdrawal bleeding. Controlled ovarian hyperstimulation (COH) was performed starting from the next menstrual cycle. In the control group, EP tablets were used for 10 days after ovulation in the cycle before the patients underwent OPU. COH was performed in the same manner as in the treatment group. In both groups, the COH protocol (long, short or antagonist protocol) was determined based on anti-Müllerian hormone level and antral follicle count. M: menstruation.

**Table 1 ijms-24-12891-t001:** Basic characteristics ^a^ of patients in the control and treated groups.

	Control (*n* = 40)	Treated (*n* = 30)	*p* Value
Age (years)	33.7 ± 4.0	33.0 ± 4.0	0.519
BMI (kg/m^2^)	21.1 ± 2.7	20.4 ± 2.3	0.256
AMH (ng/mL)	3.1 ± 2.1	2.9 ± 1.9	0.565
Unilateral endometriomas (%)	70.0 (28/40)	56.7 (17/30)	0.261
Bilateral endometriomas (%)	30.0 (12/40)	43.3 (13/30)	0.261
Largest endometrioma diameter ^b^ (mm)	29.8 ± 10.3	43.8 ± 13.0	<0.001
Antagonist protocol (*n*)	13	6	
Short protocol (*n*)	8	12	
Long protocol (*n*)	18	12	
Primary infertility (%)	70.0 (28/40)	86.7 (26/30)	0.100
Duration of infertility (months)	22.7 ± 23.0	29.0 ± 28.1	0.142
History of ovarian operation ^c^ (%)	7.5 (3/40)	10.0 (3/30)	0.712
Duration from the surgery to COH (months)	39.0 ± 12.8	44.6 ± 22.2	0.775

^a^: These parameters were obtained before DNG treatment. ^b^: The mean size of the largest endometrioma among each patient. ^c^: All ovarian surgeries were unilateral endometrioma cystectomies.

**Table 2 ijms-24-12891-t002:** Clinical outcomes of IVF–ET in the control and treated groups.

	Control (*n* = 40)	Treated (*n* = 30)	*p*
Gonadotropin (FSH) dose (IU)	2124 ± 706.2	1982 ± 480.1	0.332
Gonadotropin (FSH) duration (day)	9.5 ± 1.7	9.0 ± 1.9	0.217
Ovarian follicle ≥ 15 mm (*n*)	10.5 ± 6.4 (420)	6.6 ± 3.7 (197)	0.002
E_2_ on the trigger day (ng/mL)	2564 ± 1232	1948 ± 1072	0.027
Oocytes retrieved (*n*)	14.0 ± 9.6 (561)	10.5 ± 7.2 (316)	0.091
Mature oocytes (*n*)	11.5 ± 7.4 (459)	8.1 ± 6.2 (244)	0.046
Maturation rate (%)	81.8 (459/561)	78.0 (244/316)	0.055
Fertilized oocytes (*n*)	9.0 ± 6.6 (360)	6.5 ± 5.3 (196)	0.093
Fertilization rate (%)	78.4 (360/459)	80.3 (196/244)	0.930
Fertilization rate (IVF) (%)	71.6 (101/141)	79.4 (77/97)	0.081
Fertilization rate (ICSI) (%)	81.4 (259/318)	80.9 (119/147)	0.381
Blastocysts (*n*)	5.2 ± 4.7	3.6 ± 4.1	0.112
Blastulation rate (%)	57.8 (208/360)	54.6 (107/196)	0.751
Morphologically good blastocysts (*n*)	2.9 ± 3.1	2.3 ± 3.3	0.409
Morphologically good blastocyst rate (%)	32.5 (117/360)	34.7 (68/196)	0.620
ET cancellation rate (%)	2.5 (1/40)	10.0 (3/30)	0.182
Embryos transferred (*n*)	80	51	
Fresh embryo (*n*)	30.0 (24/80)	31.4 (16/51)	0.868
Frozen thawed embryo (*n*)	70.0 (56/80)	68.6 (35/51)	
Gestational sacs (*n*)	30	19	
Fresh embryo (*n*)	20.0 (6/30)	42.1 (8/19)	0.095
Frozen thawed embryo (*n*)	80.0 (24/30)	57.8 (11/19)	
Implantation rate (%)	37.5 (30/80)	37.3 (19/51)	0.978
Pregnancy rate (%)	70.0 (28/40)	56.7 (17/30)	0.250
Live birth rate (%)	57.5 (23/40)	40.0 (12/30)	0.147
Abortion rate (%)	20.0 (6/30)	31.6 (6/19)	0.358
Ectopic pregnancy (%)	3.3(1/30)	0 (0/19)	0.421

**Table 3 ijms-24-12891-t003:** Comparison of first and second OPU results for poor responders in the treated group ^a^.

	Treated Group (*n* = 9)		
	1st OPU ^b^	2nd OPU ^c^	*p*
Antral follicle count (*n*)	4.8 ± 2.8 (43)	8.7 ± 4.7 (78)	0.014
Gonadotropin (FSH) dose (IU)	2161 ± 510.6	2433 ± 571.3	0.135
Gonadotropin (FSH) duration (day)	9.0 ± 2.2	10.4 ± 2.4	0.056
Ovarian follicle ≥ 15 mm (*n*)	3.6 ± 1.3 (32)	7.8 ± 3.4 (70)	0.018
E_2_ on the trigger day (ng/mL)	1333 ± 552.8	2289 ± 803.0	0.034
Oocytes retrieved (*n*)	4.7 ± 2.2 (42)	9.3 ± 3.0 (84)	0.005
Mature oocytes (*n*)	3.7 ± 2.4 (33)	7.0 ± 3.9 (63)	0.038
Maturation rate (%)	78.6 (33/42)	75.0 (63/84)	0.582
Fertilized oocytes (*n*)	1.8 ± 1.7 (16)	5.6 ± 4.1 (50)	0.021
Fertilization rate (%)	48.5 (16/33)	79.4 (50/63)	0.054
Cleavage rate (%)	100 (16/16)	100 (50/50)	
Good embryos (*n*) ^d^	0	20	
Blastocysts (*n*)	7	8	
ET cancellation rate (%)	33.3 (3/9)	11.1 (1/9)	0.576
Embryos transferred (*n*)	8	15	
Gestational sacs (*n*)	2	6	
Implantation rate (%)	25.0 (2/8)	40.0 (6/15)	0.657
Pregnancy rate (%)	22.2 (2/9)	55.6 (5/9)	0.335
Live birth rate (%)	0 (0/9)	33.3 (3/9)	0.206
Abortion rate (%)	100 (2/2)	50.0 (3/6)	0.464

^a^ Of the 30 patients in the treated group, 13 patients underwent a second round of OPU, and 10 of them met the criteria for POR. Nine patients received a second OPU with COH protocol. The results of the first and second retrievals were compared among these nine patients. ^b^ The first OPU was performed on the second menstrual cycle after the completion of DNG administration. ^c^ The mean interval between the first and second OPU was 4.0 ± 1.1 months. ^d^ A good embryo was defined as an embryo that had at least five blastomeres on day 3, and <25% of its volume was filled with fragments.

## Data Availability

Not Applicable.

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
