# Peer review of "Exploring the Impact of Endometrioma Aspiration and Dienogest Combination Therapy on Cyst Size, Inflammatory Cytokines in Follicular Fluid and Fertility Outcomes"

_ijms, 2023, doi:10.3390/ijms241612891_

Round 1

Reviewer 1 Report

I was pleased to revise the manuscript entitled “Exploring the impact of aspirating endometrioma and dienogest combination therapy on cyst size, inflammatory cytokines in follicular fluid and fertility outcomes.”.

The authors performed a retrospective study aimed to assess endometrioma size reduction and outcome of IVF in patients who underwent repeated endometrioma aspiration in association with dienogest treatment compared to a control group who underwent only IVF.

In my honest opinion, the topic is interesting enough to attract the readers’ attention. This is the first case series of association of repeated endometrioma aspiration with dienogest therapy.

Nevertheless, there are different concerns about the manuscript that made me decide to accept it with major revision.

Chapter of “materials and methods” should be before chapter of “results”.

Bibliography should be intensively revised. It is not updated, some articles are not cited properly.

Considering endometrioma aspiration, only two articles are cited (16,17) while in literature there are more papers, some of them more recent than the two cited. Furthermore article 16 is cited in the introduction in the statement “This procedure is minimally invasive, simple, and helps preserve ovarian reserves”. Actually this can’t be assumed from this work, that is about a slightly different procedure than the one performed by the authors (they performed sclerotherapy with 5% Tetracycline at the end of aspiration). Moreover, a systematic review and meta-analysis was recently published on this topic, but it wasn’t included in the references.

Gao X, Zhang Y, Xu X, Lu S, Yan L. Effects of ovarian endometrioma aspiration on in vitro fertilization-intracytoplasmic sperm injection and embryo transfer outcomes: a systematic review and meta-analysis. Arch Gynecol Obstet. 2022 Jul;306(1):17-28. doi: 10.1007/s00404-021-06278-2. Epub 2021 Nov 8. PMID: 34746993.

Considering on the other hand dienogest therapy before IVF, conflicting reports are present in literature, including also a systematic review and meta-analysis not cited by the authors.

Li X, Lin J, Zhang L, Liu Y. Pretreatment of Dienogest for Women with Endometriosis in in vitro Fertilization: A Systematic Review and Meta-Analysis. Gynecol Obstet Invest. 2023;88(3):135-142. doi: 10.1159/000529400. Epub 2023 Feb 3. PMID: 36739867.

Statement “Hormonal therapies, such as the use of GnRH agonists and the progestin dienogest (DNG) administered prior to ART have also been reported to enhance clinical pregnancy rates [19–21]” should be revised according to the following Cochrane. GnRH agonists and dienogest should be considered separately.

Georgiou EX, Melo P, Baker PE, Sallam HN, Arici A, Garcia-Velasco JA, Abou-Setta AM, Becker C, Granne IE. Long-term GnRH agonist therapy before in vitro fertilisation (IVF) for improving fertility outcomes in women with endometriosis. Cochrane Database Syst Rev. 2019 Nov 20;2019(11):CD013240. doi: 10.1002/14651858.CD013240.pub2. PMID: 31747470; PMCID: PMC6867786.

They have included in the references article n.23, an article that should be excluded considering that is written in Japanese and consequently not accessible to all the readers.

Only 9 patient of 30 among treatment group underwent 2nd OPU, why? Actually not all the remaining patients got pregnant with the 1st OPU. A low response to the treatment with low number of oocyte collected is not a clear indication for a 2nd OPU. Furthermore the 2nd OPU should be compared not only with the previous one, but also with the OPU of the control group, in order to assess if the treatment proposed has benefits or not.

In conclusions, the authors’ statements cannot be assumed from the data analysis. In particular, they should demonstrate that outcomes of patients treated with endometrioma aspiration and dienogest therapy are better than controls’ in terms at least of oocytes collection and potentially of pregnancy rates. This is a necessary evidence to conclude “the importance of ensuring an adequate preparation period between the completion of the treatment and the initiation of COH for ART” considering also that endometrioma aspiration is not standard of care and is not free from risks.

The manuscript should be revised by a native English speaker in order to improve its readability.

Author Response

To Reviewer 1

We are grateful to reviewer 1 for the critical comments and useful suggestions that have helped us to improve our paper. As indicated in the responses that follow, we have taken all these comments and suggestions into account in the revised version of our paper.

Regarding: Chapter of “materials and methods” should be before chapter of “results”.

Response: Thank you for your comment. The editor of the journal instructed us to describe “introduction, results, discussion, and methods in the journal's format, so we have not made the corrections you have indicated. We appreciate your understanding.

Regarding: Bibliography should be intensively revised. It is not updated, some articles are not cited properly.

Response: Thank you for pointing out our lack of knowledge and our errors. We have followed your instructions and referred three review article stating aspiration [reference No. 18 of revised version], and articles describing that GnRHa [24], and Dienogest [25] alone do not improve ART outcomes in patients with endometriosis. More references cited [16, 17, 19, 23].

Regarding: Furthermore article 16 is cited…

Response: Thank you for pointing out our citation error. Reference 16 has been replaced with the review article you suggested [reference No. 18 of revised version].

Regarding: They have included in the references article n.23, an article that should be excluded considering that is written in Japanese and consequently not accessible to all the readers.

Response: According to reviewer’s comment, we replaced this article with English paper [reference No. 28 of revised version]. The text has also been changed due to the replacement of references at L 260-265 of revised version with changers.

Regarding: Only 9 patient of 30 among treatment group underwent 2nd OPU, why? Actually not all the remaining patients got pregnant with the 1st OPU. A low response to the treatment with low number of oocyte collected is not a clear indication for a 2nd OPU. Furthermore the 2nd OPU should be compared not only with the previous one, but also with the OPU of the control group, in order to assess if the treatment proposed has benefits or not.

Response: The situation of patients who did not have a second OPU due to pregnancy or dropout was described, and the criteria for low responder and reference to them were added. In addition, we added a description of patients who met the criteria for low response but were not included in the analysis because they used a different method of ovarian stimulation (N = 1) at L 203-209 of revised version with changes.

Regarding: In conclusions, the authors’ statements cannot be assumed from the data analysis. In particular, they should demonstrate that outcomes of patients treated with endometrioma aspiration and dienogest therapy are better than controls’ in terms at least of oocytes collection and potentially of pregnancy rates. This is a necessary evidence to conclude “the importance of ensuring an adequate preparation period between the completion of the treatment and the initiation of COH for ART” considering also that endometrioma aspiration is not standard of care and is not free from risks.

Response: We sincerely appreciate your very valuable feedback. In accordance with your suggestion, we have removed that statement and replaced it with “this study suggests a longer preparation period between the completion of the treatment and the initiation of COH for ART may improve the development of follicles” in conclusion at L 535-537 of revised version with changes. In addition, although data are not shown, no improvement was observed when comparing the second OPU results to the control.

Regarding: The manuscript should be revised by a native English speaker in order to improve its readability.

Response: According to reviewer’s comment, after the paper has been peer-reviewed, we will request English editing from IJMS's English editing service.

Reviewer 2 Report

The aim of this manuscript was to investigate the effectiveness of endometriosis aspiration combined with dienogest (DNG) treatment in improving ART outcomes in infertile patients with endometriosis, and they found that this treatment may be an effective option for shrinking endometriosis and reducing levels of inflammatory cytokines in follicular fluid. The pity of this manuscript is that the number of samples collected is insufficient, which makes it impossible to obtain differences between the two groups for some indicators. In addition, the author believes that DNG should be administered for a longer preparation period to have the potential to improve the development of follicles. However, how to define a long preparation time? Finally, I think this manuscript is worthy of reference, and I suggest that the author can propose more specific improvement methods to strengthen future clinical application.

Author Response

To Reviewer 2

We are grateful to reviewer 2 for the critical comments and useful suggestions that have helped us to improve our paper. As indicated in the responses that follow, we have taken all these comments and suggestions into account in the revised version of our paper.

Regarding: The pity of this manuscript is that the number of samples collected is insufficient, which makes it impossible to obtain differences between the two groups for some indicators.

Response: As you point out, the small sample size is a major problem in our study. However, we believe that we have found that relatively short treatment periods can reduce inflammatory cytokine levels and sustain cyst shrinkage, and that this is the cornerstone of further large-scale trials to demonstrate improved ART outcomes.

Regarding: the author believes that DNG should be administered for a longer preparation period to have the potential to improve the development of follicles

Response: We apologize for any misunderstanding caused by our poor English. What we are advocating is not to extend the duration of DNG administration, but that there should be a sufficient period of time for DNG administration to be completed and for infertility treatment to be initiated as described at L 317-320 & 334-336 of original version and L 535-537 of revised version with changes. In other words, we are talking about a longer period of preparation to start Gn administration after DNG administration has been completed and multiple spontaneous ovulations have been confirmed. To avoid any misunderstanding, we have changed the description at L 535-537 of revised version with changes, except for the English proofreading after the peer review is completed.

Regarding: the author can propose more specific improvement methods to strengthen future clinical application.

Response: We advocate a sufficient period of time between completion of DNG administration and initiation of fertility treatment, which may lead to recovery of the follicular follicle count and an increase in the number of oocytes recovered after ovarian stimulation. We describe this study suggests “a longer preparation period between the completion of the treatment and the initiation of COH for ART may improve the development of follicles” at L 535-538 of revised version with changes.

Reviewer 3 Report

The paper is very interesting and falls within the scope of this Journal.

There are some concerns:

Firstly, the order of various subparagraph must be change: Methods must be placed before results and not after discussion!

English revision by mother tongue is needed

Reference should be improved.

Recommendation: doi: 10.1016/j.ejogrb.2020.09.045 discussing the possible fertility strategies in patients with stage III-IV endometriosis.

Regarding results, I would summarize it considering the presence of tables otherwise it seems a barren shopping list.

Why did you choose aspiration instead of laser therapy?

For how many days was the antibiotic therapy continued?

Methods are clearly stated

Conclusion are in line with what is previously reported.

There are to many typing errors, please edit

Author Response

To Reviewer 3

We are grateful to reviewer 3 for the critical comments and useful suggestions that have helped us to improve our paper. As indicated in the responses that follow, we have taken all these comments and suggestions into account in the revised version of our paper.

Regarding: the order of various subparagraph must be change: Methods must be placed before results and not after discussion!

Response: Thank you for your comment. The editor of the journal instructed us to describe “introduction, results, discussion, and methods in the journal's format, so we have not made the corrections you have indicated. We appreciate your understanding.

Regarding: English revision by mother tongue is needed

Response: According to reviewer’s comment, after the paper has been peer-reviewed, we will request English editing from IJMS's English editing service.

Regarding: Reference should be improved. Recommendation: doi: 10.1016/j.ejogrb.2020.09.045 discussing the possible fertility strategies in patients with stage III-IV endometriosis.

Response: We have referred three review articles, several papers and your recommended article (doi: 10.1016/j.ejogrb.2020.09.045) at L 70 & 73-83 of revised version with revision.

Regarding: Regarding results, I would summarize it considering the presence of tables otherwise it seems a barren shopping list.

Response: We summarized the description in results section.

Regarding: Why did you choose aspiration instead of laser therapy?

Response: Laser therapy using a laparoscope allows observation of the abdominal cavity, and if adhesions are observed, they can be treated. It is also a good option for preserving ovarian reserve because it has less impact on ovarian reserve than cystectomy. We used aspiration instead of laser therapy because laser therapy requires the use of a laparoscope, while aspiration is less invasive and easier to perform because it allows transvaginal puncture.

Regarding: For how many days was the antibiotic therapy continued?

Response: Patients received preoperative 500mg ceftriaxone intravenously on the day of puncture and cefaclor 250 mg three times daily for two days starting the next day. We added description of antibiotic administration at L393-396 of revised version with changes.